# Zebrafish: A Powerful Model for Understanding the Functional Relevance of Noncoding Region Mutations in Human Genetic Diseases

**DOI:** 10.3390/biomedicines7030071

**Published:** 2019-09-16

**Authors:** Anita Mann, Shipra Bhatia

**Affiliations:** MRC Human Genetics Unit, MRC Institute of Genetics & Molecular Medicine, The University of Edinburgh, Western General Hospital, Crewe Road, Edinburgh EH4 2XU, UK; anita.mann@igmm.ed.ac.uk (A.M.)

**Keywords:** zebrafish, gene regulation, cis-regulation, human genetics

## Abstract

Determining aetiology of genetic disorders caused by damaging mutations in protein-coding genes is well established. However, understanding how mutations in the vast stretches of the noncoding genome contribute to genetic abnormalities remains a huge challenge. Cis-regulatory elements (CREs) or enhancers are an important class of noncoding elements. CREs function as the primary determinants of precise spatial and temporal regulation of their target genes during development by serving as docking sites for tissue-specific transcription factors. Although a large number of potential disease-associated CRE mutations are being identified in patients, lack of robust methods for mechanistically linking these mutations to disease phenotype is currently hampering the understanding of their roles in disease aetiology. Here, we have described the various systems available for testing the CRE potential of stretches of noncoding regions harbouring mutations implicated in human disease. We highlight advances in the field leading to the establishment of zebrafish as a powerful system for robust and cost-effective functional assays of CRE activity, enabling rapid identification of causal variants in regulatory regions and the validation of their role in disruption of appropriate gene expression.

## 1. Introduction

Human genetic diseases affect a wide range of tissues and are caused by numerous types of mutations. Understanding the cause and progression of these diseases relies heavily on the use of animals, including mouse, rat and zebrafish, to generate models mimicking the human condition. The use of these approaches for determining the aetiology of genetic disorders caused by damaging mutations in protein-coding genes is well established. However, functional analysis of mutations in the noncoding regions remains a huge challenge. Rapid technological advances have enabled the widespread application of whole-genome sequencing (WGS) for the identification of putative pathogenic mutations in patient cohorts. It has been firmly established that many of these mutations reside in the noncoding regions of the human genome, most of which are likely to harbour cis-regulatory elements (CREs) [1,2]. CRE sequences are highly enriched for binding sites of tissue-specific transcription factors (TF). The disease-associated sequence variation alters the TF binding sites in CREs, leading to aberrant CRE function and altered target gene expression [3]. Functional analysis of CRE activity, and assessment of the impact of disease-associated sequence changes on this activity, is heavily reliant on the availability of the right TFs in the right stoichiometric concentrations, which is only precisely captured in the context of animal development. Thus, although WGS has the power to identify variants in noncoding regions, the pathogenicity of these variants is much more difficult to assess compared to variants in coding regions. The use of in vivo reporter transgenic assays in mouse or zebrafish for determining their functional potential is indispensable. We describe the techniques and assays available for establishing where and when in embryonic development the enhancers act, and how this activity is affected by the presence of disease-associated mutations.

## 2. Prediction of Cis-Regulatory or Enhancer Activity in Noncoding Regions of Human Genome

Advances in genomic sequencing technologies have enabled widespread application of whole genome sequencing technology to patient DNA samples. These studies have led to the identification of clinically-relevant mutations in the noncoding regions of the human genome, most of which are likely disrupting CRE function of these sequences by altering the sequences of transcription factor binding sites [4]. The next step towards linking these sequence changes to the aetiology of the disease is defining the coordinates of CREs which harbour these sequence changes and predicting the target genes whose expression would be affected by the altered CRE function. Identifying stretches of sequence conservation between evolutionarily diverged species or duplicated gene loci using a variety of web- based tools, e.g., PIPmaker, VISTA, ECR browser, UCSC, Ensembl, etc. [5] is a useful way of prioritising elements for further functional studies. Evolutionary conservation of CRE sequence indicates functional roles of the CRE. However, these methods fail to detect evolutionarily diverged or lineage specific CREs. Furthermore, it is difficult to predict the target genes using these methods as CREs may function over large distances and may not necessarily regulate the most proximal gene [4]. A few studies have utilised conservation of sequence and synteny over large evolutionary distances as a measure of predicting putative CREs and the target genes regulated by them [6].

These predictions are further substantiated by looking for enrichments of hallmarks of enhancer function on the predicted CREs. Transcription factor profiling of the CRE sequences helps discern the possible effects of disease-associated mutations in CREs. These profiles can be predicted both computationally (using a variety of web-based tools e.g., UCSC, MEME, TRANSFAC, JASPAR, etc.) and experimentally (using ChIP-Chip, ChIP-seq, yeast one hybrid and protein binding microarrays) [7,8]. Enrichment of transcription factors, like p300 and CBP, are useful indicators of active CRE presence. Techniques like DNaseI hypersensitivity mapping by DNaseI-seq [9], FAIRE (formaldehyde-assisted isolation of regulatory elements) by FAIRE-seq [10] and Assay for Transposase-Accessible Chromatin using sequencing (ATAC-seq) [11] generate genome-wide profiles of nucleosome-free regions available for transcription factor binding, indicative of CRE presence. Histone modification profiling (in particular for H3K4Me1, H3K27Ac) by Chip-seq and ChIP-Chip [12,13] generates genome-wide enrichment profiles for active CRE-associated histone modifications, as indicators of the presence of functional CREs. Chromatin looping studies by a variety of chromosome conformation capture techniques (e.g., 3C, 4C, 5C, Hi-C, ChIA-PET), DAM-ID and 3D FISH [14,15] have been employed to detect interactions between predicted CREs and their target gene promoter, in the context of the entire gene locus.

These assays have been performed on a large number of in vitro cultured cell lines, whole embryos and tissues derived from human, mouse and zebrafish [16]. The information is publicly available on genome browsers like UCSC and Ensembl, enabling rapid analysis of predicted CREs for the presence of these features. However, since a large number of cells are required to perform these assays, they have been mostly restricted to in vitro cultured cell lines or larger tissues (containing mixed cell populations). Biological relevance of these datasets can therefore be limited due to the lack of potentially important developmental context, given the tissue-specific nature of CRE function. Recent advances in single-cell technologies [11] have the potential for overcoming this limitation, provided robust methods of obtaining the precise cell-types where the CREs are active are developed.

## 3. Testing the Predicted CRE Activity in CRE-Reporter Assays

The putative CREs harbouring the disease-associated mutations are tested in CRE-reporter assays to determine their function and how this is affected by the presence of disease-associated mutations. The most widely used assays, along with their merits and de-merits, are described in Table 1.

## 4. Zebrafish Dual-Colour CRE-Reporter Assay for Assessing Effects of Mutations on CRE Function

Although WGS has the power to identify variants in noncoding regions, the pathogenicity of these variants is much more difficult to assess compared to variants in coding regions. The use of in vivo reporter transgenic assays in mouse or zebrafish for determining their functional potential is indispensable.

Zebrafish is an excellent in vivo system for characterising putative tissue-specific CREs [19,20]. Large numbers of embryos can be injected. The resulting transgenic lines are ideal for live imaging analysis as zebrafish embryos are transparent and develop rapidly outside the mother, making it feasible to visualise or tag specific cell types in the living embryo [27]. However, the lack of sequence conservation over large portions of the noncoding parts of the human and zebrafish genomes hampered the full exploitation of this powerful system for the functional characterisation of CREs. Research over the past few years has demonstrated the potential of zebrafish models to assess the function of human and mouse CREs, irrespective of their primary sequence conservation in zebrafish [28,29,30,31,32,33]. These studies established that, in spite of significant changes in CRE sequences over the course of evolution, CREs can still capture the transcription factors required for their function in the cell and tissue types they are active in. Based on these studies, a highly robust approach was developed in our laboratory [20 (Figure 1) for testing the in vivo spatial and temporal activity of wild type and putative SNP/mutation bearing human CREs in the same developing embryo, using dual fluorescence CRE-reporter zebrafish transgenics that allow direct comparison of CRE-activity of the two alleles.

The functional output from each CRE version (wild type or SNP/mutation bearing) is visualised simultaneously as eGFP or mCherry signal within a live developing embryo bearing both transgenes. This enables unambiguous comparison of the activity of both wild type and mutant CREs in a developmental context, simultaneous assessment of multiple separate elements for subtle differences in spatio-temporal overlap, and the validation of putative TFs by analysing the effect of morpholino-mediated depletion of the putative TF on CRE activity. We established proof of principle for the robustness of the assay using known mutations associated with holoprosencephaly, identified in a CRE from the SHH locus [20]. This approach has been subsequently used for rapid functional screening of putative CRE mutations in various disease cohorts, including Aniridia [29] and Pierre Robin Sequence [31]. This assay has clear advantages over other conventional CRE-reporter transgenic assays. It provides a rapid, unambiguous detection of subtle differences in CRE activities while using a very low number of animals compared to previous single reporter assays in mouse and zebrafish (described in Table 1). Furthermore, embryos obtained from these transgenic lines could be subjected to fluorescence activated cell sorting techniques to isolate the precise cell-types where the CREs are active. These cells would serve as the ideal biological material for investigating the predicted CREs for hallmarks of CRE function by techniques described in Section 1. The current design of the assay relies on integration of the CRE-reporter constructs in the zebrafish genome mediated by Tol2 recombination sites, which are randomly distributed in the zebrafish genome [34]. CRE activities are assessed in embryos derived from multiple independent transgenic lines to rule out any bias arising in the analysis due to site of integration of the constructs. An alternate system of integrating CRE-reporter cassettes in the zebrafish genome is mediated by recombination via phiC31 recombination sites [22,23]. This allows targeted integration of the transgene into pre-defined sites in the zebrafish genome, thus removing any bias in the analysis arising from site of integration of the transgene. An assay combining the advantages of simultaneous dual-colour imaging of CRE activity of the two alleles with targeted integration of the CRE-reporter cassette in the zebrafish genome would be highly suitable for robust qualitative and quantitative assessment of the effects of disease-associated mutations on CRE activities.

The approaches described in this review enable rapid and confident identification of the stretches of noncoding DNA harbouring CREs, likely to be crucial for regulating the expression of target genes implicated in the particular disease cohort. However, since all the methods described here study the function of CREs in isolation and outside the genomic context of their native locus, these predictions would need further validation [35]. Confident assignment of pathogenicity to mutations harboured in CREs would require the use of genome-editing techniques for generating mouse or zebrafish models bearing deletion of disease-associated CRE or knock-in of the disease-associated CRE variants. The choice of the model system in these studies would be based on the CRE region being investigated. Zebrafish could be the method of choice where the CREs and the predicted target gene affected by the mutation are conserved in the zebrafish genome. Genome-editing experiments in zebrafish would also prove to be extremely powerful for genome-wide screens for identification of CREs implicated in a specific disease condition where the phenotyping assays are well-defined in zebrafish [32,33]. These studies will improve our ability to confidently and rapidly discern pathogenic vs. non-pathogenic CRE variants, and will enhance understanding of the genetics of human disorders. They will also enable confident diagnosis and genetic counselling for patients, particularly in cases where no coding region mutations in candidate genes have been identified.

## Figures and Tables

**Figure 1 biomedicines-07-00071-f001:**
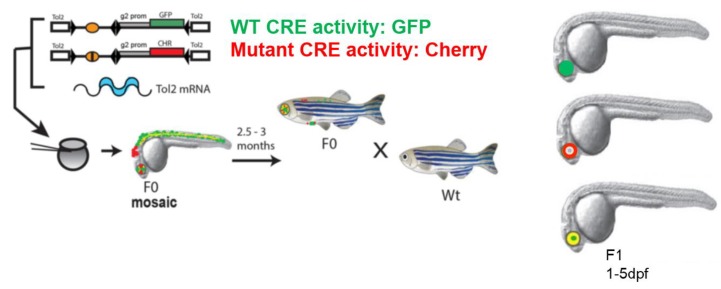
In vivo characterisation of disease-associated cis-regulatory variants by dual fluorescence reporter transgenic analysis in zebrafish. Schematic representation of the assay pipeline (modified from [20]). Both alleles (wild type and mutant) of potential disease-associated CREs are cloned in a construct with a reporter cassette of choice, to create the cis-element-reporter cassette flanked by Tol2 sites. Gata2-promoter (g2 prom) derived from mouse genome is included in the CRE-reporter constructs to serve as a minimal promoter in the assay. Co-injection of reporter constructs containing wild type (Wt) and mutant (Mut) versions of the CRE with Tol2 transposase-encoding RNA into early zebrafish embryos results in independent integration of the reporter cassettes. Transgenic founders (F0) are bred to establish transgenic lines. Expression patterns are examined in fish of F1 or later generations. Differences between Wt and Mut elements can be compared directly in the same fish using the GFP and mCherry fluorescent reporters.

**Table 1 biomedicines-07-00071-t001:** Characterisation and functional validation approaches of predicted/putative cis-regulatory elements (CREs).

**CRE-reporter assays in in vitro cultured cell lines** **Description:** CRE sequence is analysed and cloned in vectors bearing minimal promoters and reporter genes, such as luciferase or GFP High-throughput versions use oligo-based synthesis to construct large numbers of barcoded reporter genes, which are transfected into the mammalian cells and quantified by RNA sequencing of their unique barcodes **Advantages:** Suitable for high-throughput analysis (CRE-seq [17], STAR-seq [18]) Enables functional testing of thousands of different CREs and provides a better understanding of enhancer code/grammar **Limitations:** Loss of genomic context necessitating further validation in in vivo models Lacks the relevant biological context; the results cannot be extrapolated to physiological or disease conditions Bias in high-throughput methods due to stability or processing of reporter RNA
**CRE-reporter assays in zebrafish** **Description:** CRE-reporter constructs bearing CRE sequences driving fluorescent reporter genes are injected into zebrafish embryos. The spatio-temporal expression of the reporter gene is tracked by imaging as a measure of CRE activity during embryonic development, either in transiently injected embryos [19] or in embryos derived from stable transgenic lines [20] **Advantages:** Highly suitable for imaging due to transparent nature of zebrafish embryos Highly cost effective and low maintenance compared to mouse transgenic experiments Use of multiple reporter genes allows detection of qualitative comparisons of activities between two different CREs or WT vs. Mutant CRE allele [20] Some high-throughput screening platforms e.g., Automated Reporter Quantification in vivo (ARQiv) have been developed [21] CRE-reporter constructs can be integrated in pre-determined sites in the zebrafish genome using phiC31-mediated recombination [22,23] **Limitations:** Bias arises in analysis due to random integration of the CRE-reporter cassette in the zebrafish genome; multiple transgenic embryos have to be analysed per CRE for confident assignment of activity. Although this is avoided in the targeted integration strategy using phiC31 recombination, specialized zebrafish transgenic lines are needed for the assay Lack of sequence conservation over large regions of human genome and the duplication of a large number of genes in the zebrafish genome restricts the full exploitation of this powerful system Loss of genomic context, necessitating further validation in in vivo models
**CRE-reporter assays in mouse** **Description:** CRE-reporter assays in mice performed by generating stable or transient transgenic lines using constructs bearing CRE sequences driving lacZ or fluorescent reporter genes [24] **Advantages:** CRE activity analysed in vivo provides a highly relevant context of embryonic development to assess activity Results obtained can be rapidly extrapolated to human situations due to the close evolutionary distance between human and mice Whole genome views have been generated using these assays for human and mouse CREs and are available as the VISTA enhancer browser Assays adapted to capture CRE activity in the context of precise genomic location using transgenic enhancer trap lines detecting local enhancer activity at different genomic integration sites [25,26] **Limitations:** Limited application of imaging techniques for assessing CRE activity due to the opaqueness of the mice embryos Analysis of CRE activity over the complete time course of embryonic development is cumbersome, slow and expensive compared to similar analysis in zebrafish Loss of genomic context necessitating further validation in in vivo models Enhancer trap methods generate massive rearrangements in the locus investigated, thus limited application in precisely defining the region’s enhancer activity within a genomic region

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
