# Peer review of "Zebrafish: A Powerful Model for Understanding the Functional Relevance of Noncoding Region Mutations in Human Genetic Diseases"

_biomedicines, 2019, doi:10.3390/biomedicines7030071_

Round 1

Reviewer 1 Report

An introduction to CREs, their role in disease and the challenges for their validation is presented. This is followed by a brief review of some methods for functionally assessing CREs in disease leading to a focus on a method previously published by the authors.

Recommendations:

A deeper analysis of the literature would be helpful. For example, there is no mention of some important papers on enhancers in zebrafish. Some papers for consideration:

Heart enhancers with deeply conserved regulatory activity are established early in zebrafish development Xuefei Yuan, Mengyi Song, Patrick Devine, Benoit G. Bruneau, Ian C. Scott & Michael D. Wilson.  Nature Communications volume 9, Article number: 4977 (2018)

Navigating the non-coding genome in heart development and Congenital Heart Disease. Chahal G1, Tyagi S2, Ramialison M3. Differentiation. 2019 May - Jun;107:11-23. doi: 10.1016/j.diff.2019.05.001. Epub 2019 May 8

Section 1.

p.1 Line 31: The reference cited does not support the statement. Suggest amend “vast majority” to “many” or cite a supporting reference. Suggest citing original work: Science. 2012 Sep 7;337(6099):1190-5. doi: 10.1126/science.1222794 rather than review here.

Section 3 Table:

Table is a good idea and potentially very useful. It could be clearer, more concise and better referenced. Recommendation to change columns (and content) to:

Method - Description - Advantages – Limitations - References

For each advantage or limitation, it would be helpful to state the point and then follow it with brief supporting sentence. Eg.

High Throughput: Assays adapted to simultaneous analysis of large numbers of candidate CREs, etc.

Loss of genomic context:

Loss of biological context:

False positives/negatives:

Low coverage CREs not well detected: etc

“To analyse CREs in relevant biological tissues, Massively parallel functional dissection (MPFD) was used where similar strategy to MPRA was used and a 20 bp barcode was subcloned in 3 UTR of the reporter….” This section should be moved to “reporter assays in mice” section or created as a separate section as it is not in vitro cell culture assay.

In zebrafish section: the high throughput screening platforms section could be more inclusive.

Section 4.

A dual CRE-reporter assay is presented as a method to functional assess CRE activity in vivo. In the context of this review, discussion of this method in comparison to the methods reviewed would be beneficial. Specifically, it would be helpful for the reader to understand the limitations of the assay (as discussed for the other methods reviewed) as well as the advantages – for example, how differences in position and number of transgene integration sites between WT and mut transgenic lines is controlled for in order to accurately assess differences in fluorescent readout.

Recommend expansion of the concluding paragraph.

Recommend discussion or at least mention of CRISPR based methods and future potential of these for analysis of CREs in vivo, specifically in zebrafish.

General note: Overall this review is under-referenced and would benefit from more attribution. At a minimum suggest:

p.2 Line 5 Insert ref after “binding sites”

p.5 Line 7 Insert ref afterZebrafish is an excellent in vivo system for characterising putative tissue-specific CREs” and the following 2 sentences.

Author Response

An introduction to CREs, their role in disease and the challenges for their validation is presented. This is followed by a brief review of some methods for functionally assessing CREs in disease leading to a focus on a method previously published by the authors.

 Recommendations:

A deeper analysis of the literature would be helpful. For example, there is no mention of some important papers on enhancers in zebrafish. Some papers for consideration:

 Heart enhancers with deeply conserved regulatory activity are established early in zebrafish development Xuefei Yuan, Mengyi Song, Patrick Devine, Benoit G. Bruneau, Ian C. Scott & Michael D. Wilson.  Nature Communications volume 9, Article number: 4977 (2018)

 Navigating the non-coding genome in heart development and Congenital Heart Disease. Chahal G1, Tyagi S2, Ramialison M3. Differentiation. 2019 May - Jun;107:11-23. doi: 10.1016/j.diff.2019.05.001. Epub 2019 May 8

The references suggested by the reviewer have now been included in the review.

 Section 1.

p.1 Line 31: The reference cited does not support the statement. Suggest amend “vast majority” to “many” or cite a supporting reference. Suggest citing original work: Science. 2012 Sep 7;337(6099):1190-5. doi: 10.1126/science.1222794 rather than review here.

The reference suggested has now been included

 Section 3 Table:

Table is a good idea and potentially very useful. It could be clearer, more concise and better referenced. Recommendation to change columns (and content) to:

Method - Description - Advantages – Limitations - References

 For each advantage or limitation, it would be helpful to state the point and then follow it with brief supporting sentence. Eg.

High Throughput: Assays adapted to simultaneous analysis of large numbers of candidate CREs, etc.

 Loss of genomic context:

Loss of biological context:

False positives/negatives:

Low coverage CREs not well detected: etc

 “To analyse CREs in relevant biological tissues, Massively parallel functional dissection (MPFD) was used where similar strategy to MPRA was used and a 20 bp barcode was subcloned in 3 UTR of the reporter….” This section should be moved to “reporter assays in mice” section or created as a separate section as it is not in vitro cell culture assay.

 In zebrafish section: the high throughput screening platforms section could be more inclusive.

The table has been re-formatted and edited according to the suggestions of the reviewers

 Section 4.

A dual CRE-reporter assay is presented as a method to functional assess CRE activity in vivo. In the context of this review, discussion of this method in comparison to the methods reviewed would be beneficial. Specifically, it would be helpful for the reader to understand the limitations of the assay (as discussed for the other methods reviewed) as well as the advantages – for example, how differences in position and number of transgene integration sites between WT and mut transgenic lines is controlled for in order to accurately assess differences in fluorescent readout.

This section has now been edited to include the suggested points

 Recommend expansion of the concluding paragraph.

Recommend discussion or at least mention of CRISPR based methods and future potential of these for analysis of CREs in vivo, specifically in zebrafish.

The concluding paragraph has now been edited to include a role of CRISPR techniques in validating the roles of CREs in the zebrafish model

 General note: Overall this review is under-referenced and would benefit from more attribution. At a minimum suggest:

p.2 Line 5 Insert ref after “binding sites”

p.5 Line 7 Insert ref after “Zebrafish is an excellent in vivo system for characterising putative tissue-specific CREs” and the following 2 sentences.

These references have now been included

Reviewer 2 Report

A well-written and timely review on the functional characterisation of cis regulatory elements using zebrafish. The manuscript covers most of the relevant litterature and gives a comprehensive table of the advantages and disadvantages of the various systems used in Table 1. 

Major comment

My suggestion is to reorganize/rewrite part of the Table 1 to make it a little bit more crisp and easy to follow. The relevant sections in the 3 columns appear out of place in some of the sections

A previous protocol using PhiC should be also mentioned in addition to reference 24 

https://link.springer.com/protocol/10.1007%2F978-1-4939-3771-4_6

Minor comments

It is unclear where Figure 1 legend starts and ends

Zebra fish should be one word in page 1, line 26

Reference list is not according to Instructions for Authors

Author Response

A well-written and timely review on the functional characterisation of cis regulatory elements using zebrafish. The manuscript covers most of the relevant litterature and gives a comprehensive table of the advantages and disadvantages of the various systems used in Table 1. 

Major comment

My suggestion is to reorganize/rewrite part of the Table 1 to make it a little bit more crisp and easy to follow. The relevant sections in the 3 columns appear out of place in some of the sections

A previous protocol using PhiC should be also mentioned in addition to reference 24 

https://link.springer.com/protocol/10.1007%2F978-1-4939-3771-4_6

The table has been re-formatted and edited according to the suggestions of the reviewers. The suggested reference has also been included in the text

 Minor comments

It is unclear where Figure 1 legend starts and ends

Zebra fish should be one word in page 1, line 26

Reference list is not according to Instructions for Authors

These suggestions have been addressed now in the revised version of the manuscript

Reviewer 3 Report

This is a well-written short review highlighting the applicability of the zebrafish model for the functional in vivo study of cis-regulatory element activity. There are however a few limitations and small issues that need to be addressed.

The authors begin by providing a background setting on the importance of CREs followed by a number of methods on how to identify potential CRE sequences. Then they present a list of existing CRE testing strategies, and finally they focus on their CRE testing model which was recently published in PLOS One.

A major point is that the limitations of the dual color system are not indicated clearly enough. Inherent to this approach, the system does not allow for the testing of long-range effects of CRE, nor of large chromatin conformational changes. Another major question is whether the CRE-interacting factors are conserved well enough between human and zebrafish to allow a reliable interpretation of the results from the described approach. If the zebrafish ortholog does not efficiently bind the human sequence, or if no zebrafish ortholog exists for a human CRE-binding factor then the readout of the system is not valid.

Although it is only passingly referenced in the Table, I feel like the authors should also add some more discussion regarding the similarities and relevant differences between their strategy and the very similar approach that was used by Roberts et al (ref. 24) on which the dual-color system was clearly based. An important point here is that the system used by the authors is based on Tol2-based transgenesis, which obviously has the potential to create a large variability depending on the transgenic insertion site and copy number. This is something which can be avoided using the PhiC31 system. Perhaps a few lines should be added describing future possibilities to combine both systems to take advantage of the dual-color readout using the stable PhiC31 transgenesis system.

A few other points that should be addressed in the review:

-          The Massively parallel functional dissection method described in Table I (ref. 18 and 22) should be moved to the mouse in vivo tests.

-          The sentence “Although the germline transmission rate of Tol2 is 20% but the variability due to multiple and random integration is a major issue” in Table I does not make sense and should be rewritten.

-          The limitation of the study from ref. 24 described in Table I regarding the interference of the attR site is very specific to the construct used since it is inserted between the promoter and fluorescent coding sequence. This is only a minor issue which doesn’t seem like a major limitation of this method.

-          Table I: “Zebrafish are cost effective and relatively low maintenance COMPARED TO mouse…”

-          Figure 1: this picture is reproduced from the study in PLOS One, and should be referenced as such in the manuscript. The WT versus mutant CRE are not clearly indicated. The resolution is also too low.

-          The rationale for using the gata2 promoter in the construct is not clear.

Author Response

This is a well-written short review highlighting the applicability of the zebrafish model for the functional in vivo study of cis-regulatory element activity. There are however a few limitations and small issues that need to be addressed.

The authors begin by providing a background setting on the importance of CREs followed by a number of methods on how to identify potential CRE sequences. Then they present a list of existing CRE testing strategies, and finally they focus on their CRE testing model which was recently published in PLOS One.

A major point is that the limitations of the dual color system are not indicated clearly enough. Inherent to this approach, the system does not allow for the testing of long-range effects of CRE, nor of large chromatin conformational changes. Another major question is whether the CRE-interacting factors are conserved well enough between human and zebrafish to allow a reliable interpretation of the results from the described approach. If the zebrafish ortholog does not efficiently bind the human sequence, or if no zebrafish ortholog exists for a human CRE-binding factor then the readout of the system is not valid.

Section 4 has now been modified in the revised version of the review to include a more detailed discussion of the limitations of the existing sytem

Although it is only passingly referenced in the Table, I feel like the authors should also add some more discussion regarding the similarities and relevant differences between their strategy and the very similar approach that was used by Roberts et al (ref. 24) on which the dual-color system was clearly based. An important point here is that the system used by the authors is based on Tol2-based transgenesis, which obviously has the potential to create a large variability depending on the transgenic insertion site and copy number. This is something which can be avoided using the PhiC31 system. Perhaps a few lines should be added describing future possibilities to combine both systems to take advantage of the dual-color readout using the stable PhiC31 transgenesis system.

More discussion on the limitations of the dual colour sytem and comparisons with the phiC31 integrase system have now been included in the revised version or the review.

A few other points that should be addressed in the review:

-          The Massively parallel functional dissection method described in Table I (ref. 18 and 22) should be moved to the mouse in vivo tests.

-          The sentence “Although the germline transmission rate of Tol2 is 20% but the variability due to multiple and random integration is a major issue” in Table I does not make sense and should be rewritten.

-          The limitation of the study from ref. 24 described in Table I regarding the interference of the attR site is very specific to the construct used since it is inserted between the promoter and fluorescent coding sequence. This is only a minor issue which doesn’t seem like a major limitation of this method.

-          Table I: “Zebrafish are cost effective and relatively low maintenance COMPARED TO mouse…”

The table has been re-formatted and edited according to the suggestions of the reviewers

-          Figure 1: this picture is reproduced from the study in PLOS One, and should be referenced as such in the manuscript. The WT versus mutant CRE are not clearly indicated. The resolution is also too low.

The reference is now included in the figure legend and a high resolution image included in the revised submission

-          The rationale for using the gata2 promoter in the construct is not clear.

This has now been included in the revised version of the manuscript in section 4